# Genome-Wide Identification and Characterization of the Abiotic-Stress-Responsive LACS Gene Family in Soybean (*Glycine max*)

**Jie Wang, Xiaoxue Li, Xunchao Zhao, Chen Na, Hongliang Liu, Huanran Miao, Jinghang Zhou, Jialei Xiao, Xue Zhao \* and Yingpeng Han \***

Key Laboratory of Soybean Biology in Chinese Education Ministry, Northeast Agricultural University, Harbin 150030, China; w15536351237@163.com (J.W.); lxx15103733950@163.com (X.L.); zhaoxunchao2017@163.com (X.Z.); 15776502797@163.com (C.N.); l974840165@163.com (H.L.); 15648005556@163.com (H.M.); zhoujh0030@163.com (J.Z.); j_l_x@163.com (J.X.)
**\*** Correspondence: xuezhao@neau.edu.cn (X.Z.); hyp234286@aliyun.com (Y.H.); Tel.: +86-451-5519-0778 (Y.H.)

**Abstract:** Long-chain acyl-CoA synthases (LACSs) are a key factor in the formation of acyl-CoA after fatty acid hydrolysis and play an important role in plant stress resistance. This gene family has not been research in soybeans. In this study, the soybean (*Glycine max* (L.) Merr.) whole genome was identified, the LACS family genes of soybean were screened, and the bioinformatics, tissue expression, abiotic stress, drought stress and co-expression of transcription factors of the gene family were analyzed to preliminarily clarify the function of the LACS family of soybean. A total of 17 LACS genes were screened from soybean genome sequencing data. A bioinformatics analysis of the GmLACS gene was carried out from the aspects of phylogeny, gene structure, conserved sequence and promoter homeopathic element. The transcription spectra of GmLACSs in different organs and abiotic stresses were used by qRT-PCR. The GmLACS genes, which co-expresses the significant response of the analysis of drought stress and transcription factors. The results showed that all soybean LACS have highly conserved AMP-binding domains, and all soybean LACS genes were divided into 6 subfamilies. Transcriptome analysis indicated that the gene-encoding expression profiles under alkali, low temperature, and drought stress. The expression of GmLACS9/15/17 were significantly upregulated under alkali, low temperature and drought stress. Co-expression analysis showed that there was a close correlation between transcription factors and genes that significantly responded to LACS under drought stress. These results provide a theoretical and empirical basis for clarifying the function of LACS family genes and abiotic stress response mechanism of soybean.

**Keywords:** soybean (*Glycine max* (L.) Merr.); long-chain acyl-CoA synthases (LACSs); bioinformatics; expression analysis; drought stress; resistance

## 1. Introduction

Fatty acids (FAs) are an important component of oil. It is widely distributed in plant cells and is essential for plant growth and development. In plants, FAs are combined to from phospholipids, membrane glycerides, sphingolipids, and triacylglycerol (TAGs), which become the main energy storage mode in plants [1–4]. It can also be the precursor of cuticle, cork fat and surface wax, as a surface barrier against biological and abiotic stresses [5–7]. The anabolic pathway of plant fatty acids is mainly accomplished in a variety of organelles, involving synthesis in plastids, elongation in endoplasmic reticulum (ER), and β-oxidative decomposition in peroxisomes [5,8,9]. In order to complete the process of "entering the cell membrane, transporting between different organelles and leaving the organelles", fatty acids usually release free fatty acids through water interpretation, which are activated by long chain acyl-CoA synthases (LACSs) to form acyl-CoA, and then

generate TAG in the endoplasmic reticulum [10–12]. Therefore, LACS played an important role in the FAs anabolic pathway [13,14].

FAs can be divided into three categories according to the length of their carbon chains, namely long-chain fatty acids (LCFA), medium chain fatty acids (MCFA) and short chain fatty acids (SCFA). The length of fatty acids in higher plants is usually between 14–20 carbon long-chain fatty acids [15]. Acyl CoA synthase (ACS) is also known as fatty acid CoA ligase. According to the difference of carbon chain length of specific fatty acid substrates, ACS can be divided into the following four categories: super long chain (>C20), long chain (C14–C18), medium chain (C10–C12) and short chain (C6–C8) acyl CoA synthetases [16].

Long-chain acyl-CoA synthase has a highly conserved AMP-binding domain [17,18]. *LACS* preferentially activate long-chain FAs (LCFAs; C16, C18) formation of acyl-CoA [17,19]. It is considered to be an intercellular fatty acid transporter and a key step necessary for the utilization of fatty acids in plant metabolism [3,20,21]. *LACS* catalyzed the formation of acyl-CoA mainly through two-step reaction, free fatty acids were broken down into adenosine acid, binding to CoA to release AMP and acyl-CoA [8,13]. The spatial distribution of LACS enzymes in cells is a factor that leads fatty acids to a specific metabolic fate [22,23]. Consistent with this, in most eukaryotes *LACS* is encoded by different gene subfamilies in specific pathways, such as tissue-specific expression and subcellular location. However, LACS activity often shows significant overlap in substrate specificity, such as human fatty acid translocation, and this is also the case in Arabidopsis thaliana [20,23].

In *Arabidopsis thaliana*, nine *LACS* isoforms were identified, which had different expression patterns and functions. In vitro enzyme activity analysis showed that all LACSs can effectively activate a variety of substrates [24]. Meanwhile, most of the nine LACS genes in *Arabidopsis* have been isolated and mutant. The identification of these mutants and the analysis of subcellular localization of expression patterns revealed a complex LACS activity network involving different aspects of lipid metabolism. Among them, several LACS subtypes located in the endoplasmic reticulum can activate fatty acids to produce surface lipids. Long chain specificity analysis and the phenotype of *lacs1* mutant showed that LACS1 played a major role in the production of long chain acyl-CoA, and LACS1 was the precursor of cuticle wax. Together with LACS1, LACS2 activates VLCFAs to produce wax components and to bind to keratin in C16 and C18 acyl groups [17,20]. LACS4 and LACS1 are partially redundant in providing substrates for wax biosynthesis in stem and leaf cuticle and lipid formation in pollen coat [21]. LACS3 may be strongly expressed in stem epidermis, but it has not been studied so far [22]. LACS5 is expressed in anthers. Similarly, the identification of *lacs6* and *lacs7* double mutants showed that, *LACS6* and *LACS7* were involved in the degradation of fatty acids in peroxidase [25]. In addition, *LACS9* was considered to be the main *LACS* subtype involved in the formation of acyl-CoA. Participate in TAG biosynthesis [17,18,20]. LACS9 and LACS4 overlap with LACS8 in function. The destruction of *LACS8* under the background of *lacs9* and *lacs4* may lead to lethality [21]. LACS9 was a widely studied member of the *AtLACS* family and was considered to be a major LACS subtype due to its location in the de novo and synthetic plastids of plant fatty acids. Additionally, LACS genes have been found in many higher plants including cash crops, such as rice, corn, cotton, castor, goat grass, rice, apple, and brassica napus [26–31]. The above results show that LACS gene plays an important role in lipid synthesis, catabolism, stress resistance and yield of cash crops. Although the biological functions of LACS have been described in some of the model plants mentioned above, no relevant study on the *GmLACS* gene family has been reported.

Soybean was a widely cultivated oil crop in the world [32]. The high-quality fatty acid components of soybean determined edible oil and its nutritional value [33]. China was one of the main production areas of soybean, and with the continuous increase in population, the demand for soybean was also increasing. Studies have shown that the *LACS* gene family plays an irreplaceable role in lipid synthesis and catabolism in plants, which is of great significance for improving lipid yield in plants. Therefore, the study of soybean *LACS* gene family was very urgent and important. Based on the whole genome sequencing data of

soybean, this study used bioinformatics methods to identify the whole genome of soybean LACS gene family, screened the genes of soybean LACS gene, and analyzed the physical and chemical properties, phylogeny, gene structure, tissue expression, conserved motifs, abiotic stress and co-expression of the gene family. In this study, 17 members of soybean LACS gene family (GmLACS1-17) were screened. It was noteworthy that *GmLACS9* and *GmLACS17* showed significant transcriptional responses to drought stress. These results provide reference for future studies on *LACS* gene function in soybean.

## 2. Materials and Methods

### 2.1. Identification of the LACS Gene Family in Soybean

In order to obtain all *LACSs* from the whole soybean genome, a systematic BLASTP search was performed on the Phytozome database (https://phytozome-next.jgi.doe.gov/) (accessed on 7 October 2021) using published *Arabidopsis LACSs* as the query object. The protein sequences of putative soybean *LACS* family members with an E-value ($<10^{-10}$) and a sequence identity threshold > 90% were downloaded. According to the protein sequence of the screened gene, query in the genomic mode of the domain database (SMART) (http://smart.embl-heidelberg.de/) (accessed on 15 October 2021) to show whether its domain is AMP-binding domain (PF00501) to determine whether it is a candidate gene [34]. EMBL-PFAM (http://pfam.xfam.org/) (accessed on 15 October 2021) is a database of protein families, including annotations and multiple sequence alignments generated using hidden Markov models. Enter PF00501 into PFAM-A sub library, and the query results show that its domain belongs to LACS family. It is further judged that the candidate genes belong to LACSfamily genes [35]. IBS 2.0 [36] software was used to visualize the domain distribution. All *LACSs* were obtained in the same manner in the whole soybean genome.

### 2.2. Analysis of Gene Sequence and Physicochemical Properties

The identification of the *LACS* gene family in soybean. The physical and the chemical properties, including molecular formula, molecular weight, and isoelectric point, were obtained from the ExPASy website (https://www.expasy.org/) (accessed on 18 October 2021) [37]. SOPMA (https://npsa-prabi.ibcp.fr/cgi) (accessed on 18 October 2021) predicts that encodes secondary structures of proteins. And the peptide was analyzed by SignalP 4.1 (https://services.healthtech.dtu.dk/service.php?SignalP-4.1) (accessed on 18 October 2021). Protein trans-membrane regions were predicted by TMHMM Server.V.2.0 (https://services.healthtech.dtu.dk/service.php?TMHMM-2.0) (accessed on 18 October 2021).

### 2.3. Phylogenetic Tree Construction and Protein Conserved Motif Analysis

The full-length LACSs amino acid sequences of soybean (*Glycine max*), *Arabidopsis* (*Arabidopsis thaliana*), maize (*Zea mays*) and upland cotton (*Gossypium hirsutum*) [38] were obtained from Phytozome database. The multiple sequence alignments of amino acid sequences of the LACS were performed using MEGA 5.0 software, and the phylogenetic trees were constructed separately for the LACS using the Neighbor-Joining (NJ) method with the bootstrap values set at 1000 replicates [39].

We took of MEME Version 5.1.1 (http://meme-Suite.Org/tools/meme) (accessed on 4 November 2021) for identification of protein conserved motifs. The maximum number of motifs was 10, and the optimal motifs width was limited between 6 and 100 residues [40]. TBtools software was used for visualization [41].

### 2.4. Gene Location on Chromosome and Gene Structure of GmLACSs

The chromosome location data of soybean were downloaded from Phytozome database, and the *LACS* gene location on soybean chromosomes was analyzed by MG2C (http://mg2c.iask.in/mg2c_v2.1/) (accessed on 4 November 2021). The exon/intron structures of *GmLACSs* were unveiled at the GSDS (http://gsds.cbi.pku.edu.cn/index.php) (accessed on 13 November 2021) [42]. The position information of soybean AMP-binding domain

was downloaded from Phytozome database. IBS 2.0 software was used to visualize the domain distribution.

### 2.5. Collinearity Analysis of GmLACSs

According to the Ensembl Plants database (https://plants.ensembl.org/index.html) (accessed on 27 October 2021), the DNA sequences and annotated file GFF3 of soybean, Arabidopsis, maize, and upland cotton are found, and the linear relationships within soybean species and between soybean and other species were generated by using TBtools software (TBtools_windows-x64_1_098667; Chengjie Chen; China).

### 2.6. Promoter Analysis of GmLACSs

The soybean genome sequence data were downloaded from the Phytozome database, and the 2.0 kb upstream of the start codon of *GmLACS* gene was intercepted. The plant CARE (http://bioinformatics.psb.ugent.be/webtools/plantcare/html/) (accessed on 6 November 2021) was used to analyze cis-acting elements related to plant growth and development, plant hormones, and abiotic and biotic stresses.

### 2.7. Expression Profiles of GmLACSs in Diverse Tissues

The expression data of soybean *GmLACSs* at different tissues and developmental stages were acquired from the Phytozome database. The heat maps representing the gene expression intensities were generated, and cluster analysis was completed by TBtools software (TBtools_windows-x64_1_098667; Chengjie Chen; China).

### 2.8. Quantitative Real-Time RT-PCR Analysis

To analyze transcriptional profiles of *GmLACSs* during development, total RNA was extracted from soybean seeds at different development stages (10, 20, 30, 40 days, DAF). The expression level of *GmLACSs* in developing seeds at 10 DAF was used as a calibrator. To examine the transcriptional profiling of *GmLACSs* under various abiotic stresses, soybean seedlings at the second trifoliolate stage were subjected to alkali stress induced by 100 mM $NaHCO_3$, low temperature stress induce by 4 °C, and osmotic stress induce by 20% ($w/v$) PEG (with a molecular weight of 6000 g/M). Total RNA was extracted from leaf samples at 0, 6, 12 and 24 h after the above treatments. The transcripts of *GmLACSs* in soybean leaf under normal environment condition were used as a calibrator. Each quantitative real time-polymerase chain reaction (qRT-PCR) reaction was performed in triplicate (technical replicates) on three biological replicates and the transcriptional level of *GmLACSs* was calculated based on the $2^{-\Delta\Delta ct}$ method, and all of the primers used for qRT-PCR analysis were shown in Supplementary Table S1.

### 2.9. Co-Expression of GmLACSs and Transcription Factors under Drought Stresses

All of the transcription factors for drought stress response were screened in the Plant Transcription Factor Database (http://planttfdb.gao-lab.org/) (accessed on 26 February 2022) in combination with drought transcriptome data. (GSE57252) [43]. We named the strong drought response genes as guide genes and explored their co-expression relationship with corresponding transcription factors. Correlation coefficients were calculated from the expression data of guide genes and transcription factors. When Pearson correlation was higher than 0.90, co-expression network was constructed. Using Cytoscape to achieve co-expression network visualization.

### 2.10. Statistical Analysis

All of the experiments were performed with at least three biological replicates. Values are presented as mean $\pm$ SD. The significance of the data was evaluated using Student's *t*-test with SPSS statistics 22.0 software. The significance level was set at $p < 0.05$ and $p < 0.01$.

## 3. Results

### 3.1. Identification Sequence Analysis and Physicochemical Properties Analysis of LACS Gene Family in Soybean

Based on 9 *AtLACS* genes (*AtLACS1-9*) in *Arabidopsis*, BLASTP was used to search for soybean LACS protein sequences in Phytozome database, and then the redundant sequences were compared. Pfam and SMART domain search were used to verify whether *LACS* candidate genes contained AMP-binding domain. Finally, a total of 17 *GmLACS* candidate genes were identified in soybean genome, which were named *GmLACS1-17* (Table 1) according to domain information and location in the genome. Different from the 19 *GmLACS* candidate genes identified by Ayaz et al. [44], the 3 candidate genes (*Glyma.10G249700*, *Glyma.15G220900*, *Glyma.20G143900*) containing other domains outside the AMP-binding domain were removed. A newly identified candidate gene (*Glyma.20G060100*) was added. Similarly, 9 *ZmLACS* (*ZmLACS1-9*) and 17 *GhLACS* (*GhLACS1-17*) candidate genes were identified in genome (Supplementary Table S2) according to domain information and location in genome. We compared this with the protein sequences of *GmLACS*, *AtLACS*, *ZmLACS* and *GhLACSs* to remove redundancy.

**Table 1.** Basic information of the seventeen soybean *LACS* genes (*GmLACSs*).

| Gene Name | Gene ID | CDS Length (bp) | Protein Length (aa) | Isoelectric Point (pI) | Molecular Weight (Da) |
|---|---|---|---|---|---|
| *GmLACS1* | Glyma.01G225200 | 2016 | 672 | 6.88 | 75,365.95 |
| *GmLACS2* | Glyma.02G010300 | 1983 | 661 | 6.54 | 74,382.04 |
| *GmLACS3* | Glyma.03G221400 | 1989 | 663 | 6.78 | 74,409.94 |
| *GmLACS4* | Glyma.05G216600 | 1998 | 666 | 6.10 | 74,670.93 |
| *GmLACS5* | Glyma.06G112900 | 2085 | 695 | 8.24 | 76,017.50 |
| *GmLACS6* | Glyma.07G161900 | 1983 | 661 | 6.20 | 73,852.74 |
| *GmLACS7* | Glyma.10G010800 | 1983 | 661 | 6.73 | 74,516.20 |
| *GmLACS8* | Glyma.11G017900 | 1992 | 664 | 7.76 | 74,031.44 |
| *GmLACS9* | Glyma.11G122500 | 1971 | 657 | 6.23 | 73,788.63 |
| *GmLACS10* | Glyma.12G047400 | 1971 | 657 | 5.58 | 73,843.48 |
| *GmLACS11* | Glyma.13G010100 | 2178 | 726 | 7.83 | 79,550.53 |
| *GmLACS12* | Glyma.13G079900 | 2091 | 697 | 5.91 | 76,131.58 |
| *GmLACS13* | Glyma.14G149700 | 1944 | 648 | 5.90 | 70,974.48 |
| *GmLACS14* | Glyma.19G218300 | 1989 | 663 | 6.50 | 74,643.20 |
| *GmLACS15* | Glyma.20G007900 | 1983 | 661 | 6.39 | 73,831.71 |
| *GmLACS16* | Glyma.20G060100 | 2178 | 726 | 6.87 | 79,700.80 |
| *GmLACS17* | Glyma.20G060300 | 2025 | 675 | 6.53 | 74,317.36 |

All of the *GmLACS* gene sequences were analyzed in Phytozome database and Expasy database for structural analysis and prediction of physicochemical properties of protein sequences. The results showed (Table 1) that the CDS sequence length of *GmLACS1-17* was 1944–2178 bp, and the amino acid length was 648–726 aa. The isoelectric point of the proteins ranged from 5.58–8.24, and the relative molecular weight of the protein molecule ranged from 70,974.48–79,550.53 Da, suggesting that most of the proteins were neutral. In the total mean hydrophobic index, 17 genes were all negative hydrophilic proteins (Supplementary Table S3).

SOPMA predicted the protein secondary structure of *GmLACS* gene family, and the results showed (Supplementary Table S3) that the protein secondary structure of GmLACS family members consisted of α-helix, β-rotation, extended chain and random curl. Online SignalP 5.0 analysis found that 17 members of *GmLACS* family had no signal peptide characteristics. The online website TMHMM Server V. 2.0 found that *GmLACS11/16* had transmembrane structures, while none of the other family members had transmembrane structures (Supplementary Table S3). It was speculated that *GmLACS11/16* were membrane proteins, and all of the other family members were non-membrane proteins.

### 3.2. Phylogenetic, Protein Domain and Protein Conserved Motifs Analysis of GmLACSs

The full-length LACSs amino acid sequences of soybean (*Glycine max*, GmLACSs), *Arabidopsis* (*Arabidopsis thaliana*, AtLACSs), maize (*Zea mays*, ZmLACSs) and upland cotton (*Gossypium hirsutum*, GhLACSs) were obtained from Phytozome database and were used to construct the phylogenetic tree of LACSs. As shown in Figure 1A, the GmLACS protein family was divided into six subfamilies (Cluster i–Cluster vi); then, they were placed in different branches of Arabidopsis thaliana, maize and upland cotton, and each branch contained different numbers of genes. AtLACS1 was a subbranch with GmLACS2/3/7/14, and AtLACS2 branch contains GmLACS6/9/10/15. AtLACS3-5 with the distribution of GmLACS1/4/8. AtLACS8 is subbranch of GmLACS11/16/17. AtLACS9 was a branch of GmLACS5/12/13. GmLACSs was not present in particular AtLACS6-7 branches. Significantly, LACS homologous genes were significantly amplified in soybean compared with Arabidopsis.

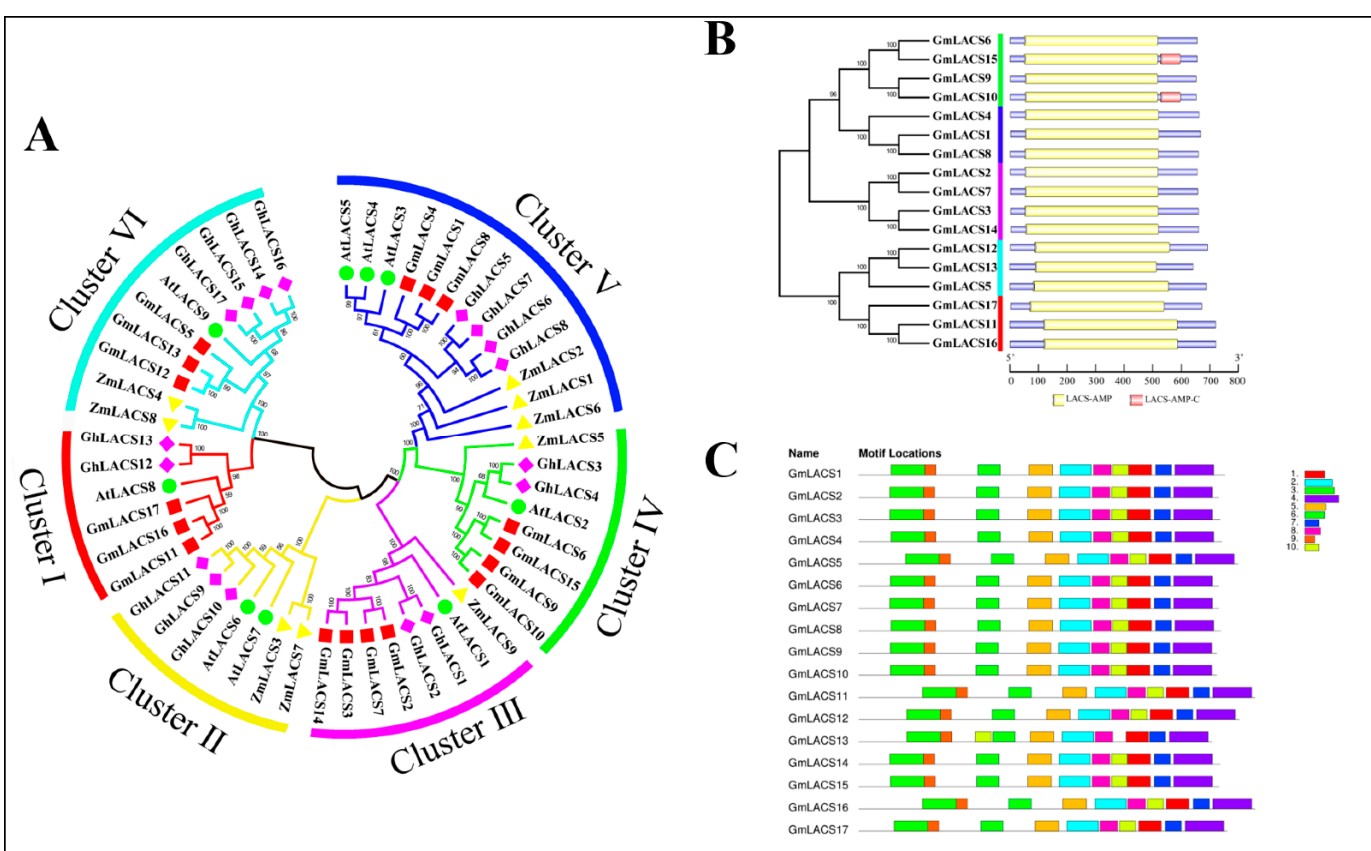

**Figure 1.** Phylogenetic, protein domain and protein conserved motifs analysis of GmLACSs. (**A**) Phylogenetic tree of LACS proteins from GmLACSs (red squares), AtLACSs (green circles), ZmLACSs (yellow triangles), and GhLACSs (purple diamonds). (**B**) The protein domain was sequenced and analyzed by subfamily. The blue circle was the protein sequence length (5′-3′), the yellow circle was GmLACS-AMP, and the red circle was GmLACS-AMP-C. Each protein sequence is proportional in length. (**C**) The conserved motifs were identified by MEME program. Each motif is indicated by a colored box that displays a number and gray lines represent the non-conserved sequences. The length of motifs in each protein is proportional.

The protein domain of GmLACSs was visually analyzed according to the classification order of 6 subfamilies, and the result was shown in Figure 1B. GmLACSs only contained AMP-binding domain (PF00501), and the AMP-binding domain was in the same position in the same branch. The positions of AMP-binding domains in different branches were basically the same, which proved that GmLACSs protein was highly conserved. We also used MEME to predict 17 GmLACSs protein conserved motifs, the number of conserved

mods in each subfamily was basically consistent with the classification. As shown in Figure 1C, all 17 GmLACS protein family members contained 10 conserved motifs, and the number and category of conserved motifs of different subfamily members were basically consistent, with little difference, which may be related to the same location of each subfamily member in the cell, that is, GmLACS protein is highly conserved.

### 3.3. Gene Location Analysis and Gene Structure Analysis of GmLACSs

The chromosome localization of 17 *GmLACS* gene family members was analyzed based on soybean genome data and MG2C website. According to the analysis results (Supplementary Figure S1), *GmLACS* genes were unevenly distributed on 13 chromosomes, among which 20 chromosomes were the most distributed (3 genes, *GmLACS15-GmLACS17*). To understand the diversity of *GmLACS* gene structure, we compared the localization and size of exons and introns in *GmLACS*. As shown in Figure 2, most *GmLACS* genes have similar gene structure, which consists of coding region and non-specific coding region. The number of exons ranged from 11–19, and there was no intron deletion. Some *GmLACS* genes have only a coding region, and all of them have AMP-binding domain. Genes in the same branch have similar exon and intron structure and number of exons. For example, 4 genes, *GmLACS6/9/10/15* are located in the same branch with 19 exons and basically the same length of exons. These results further indicate that *GmLACS* gene is highly conserved in the gene sequence and the exon intron structure.

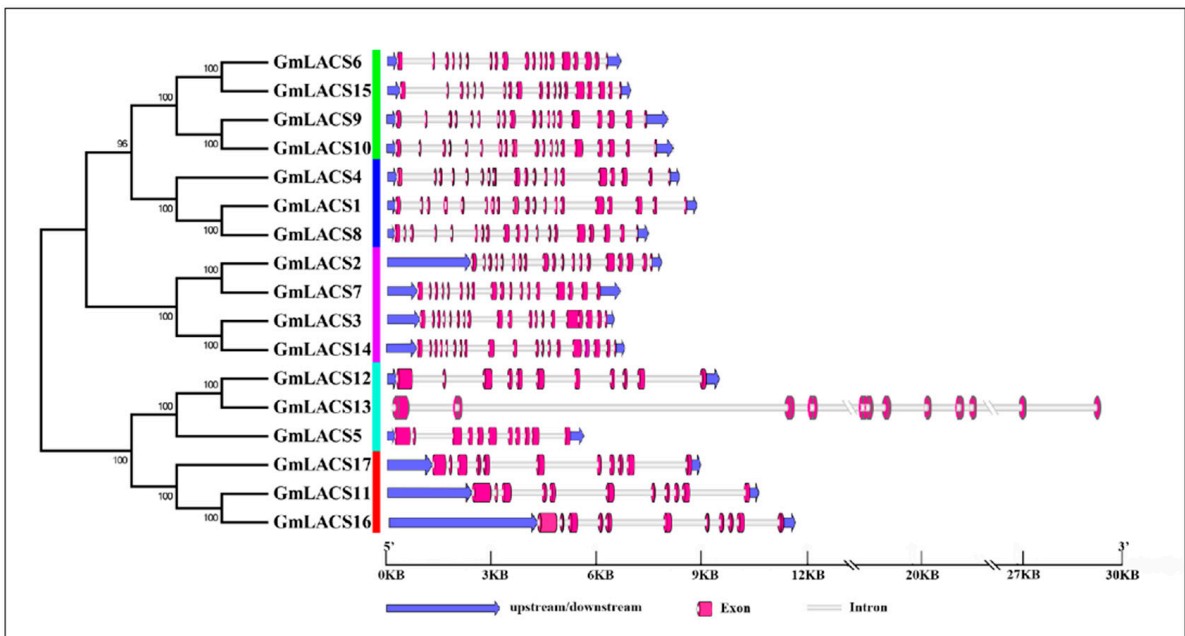

**Figure 2.** Gene structure analysis of *GmLACSs*. The gene structure was analyzed by sequencing according to subfamilies. The blue arrow was the non-coding region, 5′-3′ in the direction of the arrow, the red circle was the exon, and the gray line was the intron. The length of each gene sequence is proportional.

### 3.4. Collinearity Relation of GmLACSs

To further understand the potential function of *GmLACSs*, an intraspecific homolinearity analysis was performed on 17 *GmLACSs*. As shown in the Figure 3A, 17 GmLACSs were distributed on 13 out of 20 soybean chromosomes, and each chromosome was composed of 1-3 *GmLACS*. A total of 20 *GmLACS* homologous gene pairs were found (Supplementary Table S4), among which the *GmLACS* gene on chromosome 11 had more homologous gene pairs. Besides, the interspecies homolinearity of soybean, *Arabidopsis*, maize and upland cotton was analyzed. Based on the relative distance between species, we paired the four species and compared the collinearity of each combination to determine the species order-

ing problem. Finally, we analyzed the collinearity of species in the order of Arabidopsis, soybean, upland cotton and maize. As shown in the Figure 3B, there were many homologous gene pairs between *Arabidopsis* and soybean. In addition to the fact that *Arabidopsis* was a model plant and there are more and more in-depth studies on Arabidopsis, all the LACS genes of Arabidopsis have been excavated, 17 GmLACSs were derived from BLASP based on AtLACSs, so there were many homologous gene pairs. The results also showed that there were the most homologous gene pairs between soybean and upland cotton, but fewer homologous gene pairs between upland cotton and maize.

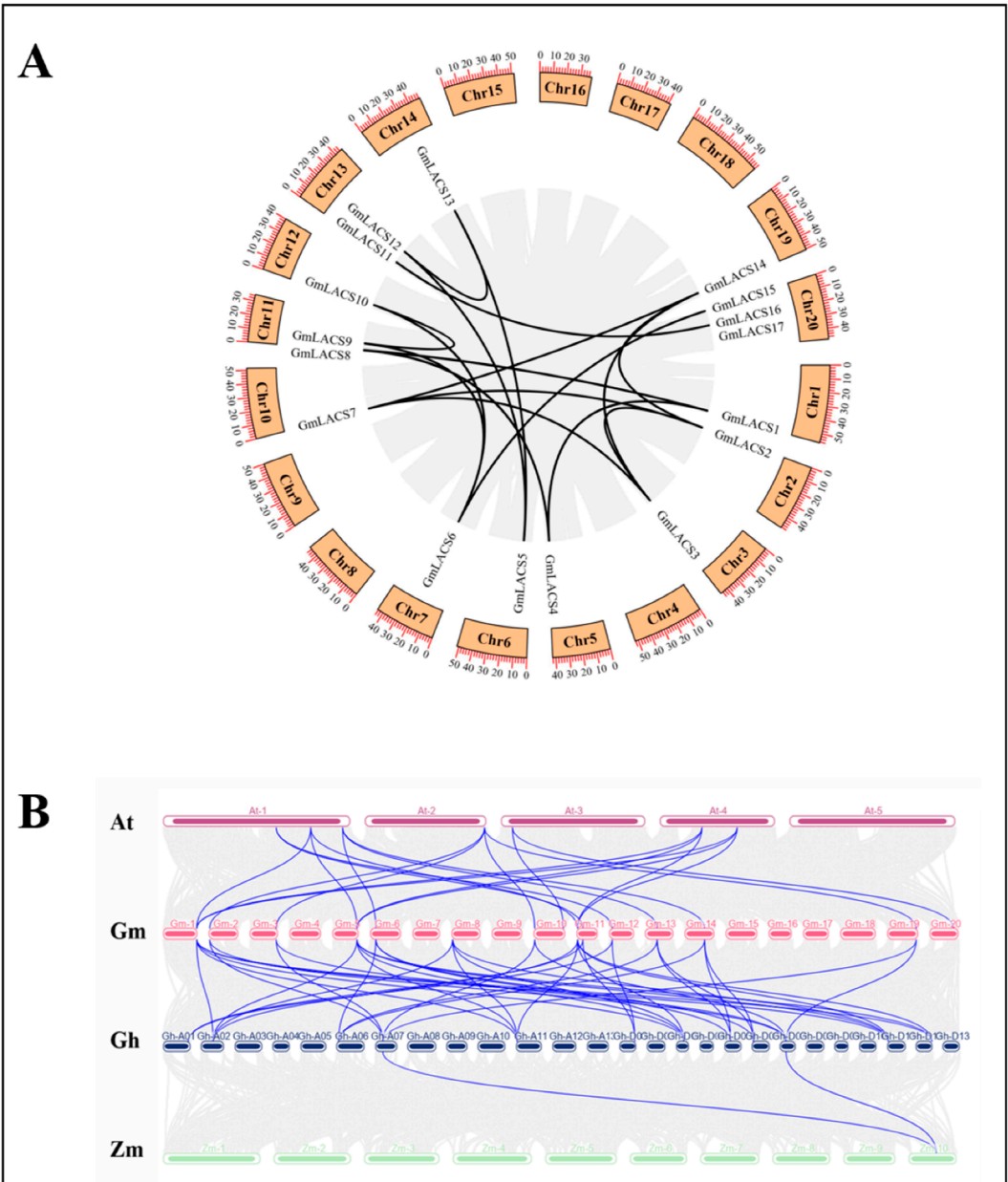

**Figure 3.** Collinearity analysis of genes in GmLACS family. (**A**) Collinearity analysis of 17 *GmLACS* genes. The soybean chromosome is shown as an orange arc, the red scale shows chromosome length in Mb, the black highlighted curve shows the *GmLACS* collinearity region, and the gray curve shows the genome-wide collinearity of soybean. (**B**) Collinearity analysis of *GmLACS*, *AtLACS*, *GhLACS* and *ZmLACS*. From top to bottom, the chromosomes of these species are Arabidopsis, soybean, upland cotton, and maize. The blue highlighted curve represents the collinearity region of *LACS*, and the gray curve represents the genome-wide collinearity of the above-mentioned species.

### 3.5. Identification of Cis-Acting Regulatory Elements in the Promoter of GmLACS Genes

To explore the regulation of *GmLACS* family members, analysis of the cis-acting regulatory elements in the 2 kb region upstream of the initiation codon of all *GmLACS* family members was conducted using the PlantCARE online portal. We identified 23 cis-acting elements related to plant growth and development, plant hormones, abiotic and biological stress. As shown in the Figure 4A, among them, there were five kinds of elements related to plant growth and development, which were endosperm expression (GCN4-Motif) [45], the gliadin metabolic ($O_2$-site), meristem expression and specific activation elements (CCGTCC-box and CTA-box), the regulation of circadian rhythms (circadian) [46]. There were eleven components associated with plant hormones; they included: STRE, methyl jasmonate response elements (CGTCA-motif and TGACG-motif) [47], ABA response element (ABRE) [48], gibberellin response elements (GARE-motif and P-box) [49], auxin response elements (TGA- element and AuxRR-core) [50], ERE, the SA response element (TCA-element) [51], TATC-box. There were 7 elements related to abiotic and biological stress, which were: ARE, LTR, MBS, WRE3, TC-rich repeats, GC-Motif, mechanical injury response (WUN-motif). Notably, the components involved in circadian rhythm regulation were detected only in the promoter region of *GmLACS4*, the regulation of ERE elements were detected only in the promoter region of *GmLACS15*. Regulation of TATC-Box and ARE elements was detected in all of the *GmLACS* promoter sequences, regulation of the STRE, ABRE, MBS, and WRE3 components was detected in almost all GmLACS promoter sequences with few exceptions. A large number of biotic and abiotic stress-related elements are found in the *GmLACS* promoter sequence, indicating that these elements play a crucial role in regulating the function of *GmLACS* gene in plant growth and development (Figure 4B).

### 3.6. Expression Profiles of GmLACSs in Different Tissues and Developmental Phases

Using the transcription patterns of *GmLACSs* in multiple tissues in Phytozome database, high-throughput sequencing data including flower, leaves, nodules, pod, root, root hairs, seed, shoot apical meristem, stem were analyzed. As shown in the Figure 5A, the transcripts of seventeen *GmLACSs* could be observed in all the tissues tested, but the expression pattern of *GmLACS* gene was significantly different in different tissues or developmental stages. In general, the expression level of *GmLACSs* in vegetative growth stage was lower than that in reproductive growth stage. Among them, the transcription level of the *GmLACS1* was relatively high in all 9 tissues, the *GmLACS15* was strongly expressed in roots, and *GmLACS12* was strongly expressed in leaves. The transcription levels of *GmLACS3/13/14* genes in 9 tissues were relatively low. The tissue-specific expression characteristics of these genes reflect their multifunctional characteristics in many aspects of soybean growth and development and prove that these genes play an important role in plant morphogenesis.

Further experiments, in order to explore the potential role of *GmLACS* gene in soybean seed development, we further confirmed the transcription patterns of *GmLACS* gene at different stages of soybean seed development by qRT-PCR (Figure 5B). All of the tested genes had different expression levels at different stages of seed development (10, 20, 30, and 40 DAF). *GmLACS1/5/9/12* maintained high expression levels at the later stages of seed development (30, 40 DAF). *GmLACS8/10/11/16/17* maintained high expression levels in the early stage of seed development (20 DAF). *GmLACS4* maintained a low level during the whole process of seed development. The transcriptional abundance of *GmLACS1* gene was highest at several stages of seed development (10, 20, 30, 40 DAF).

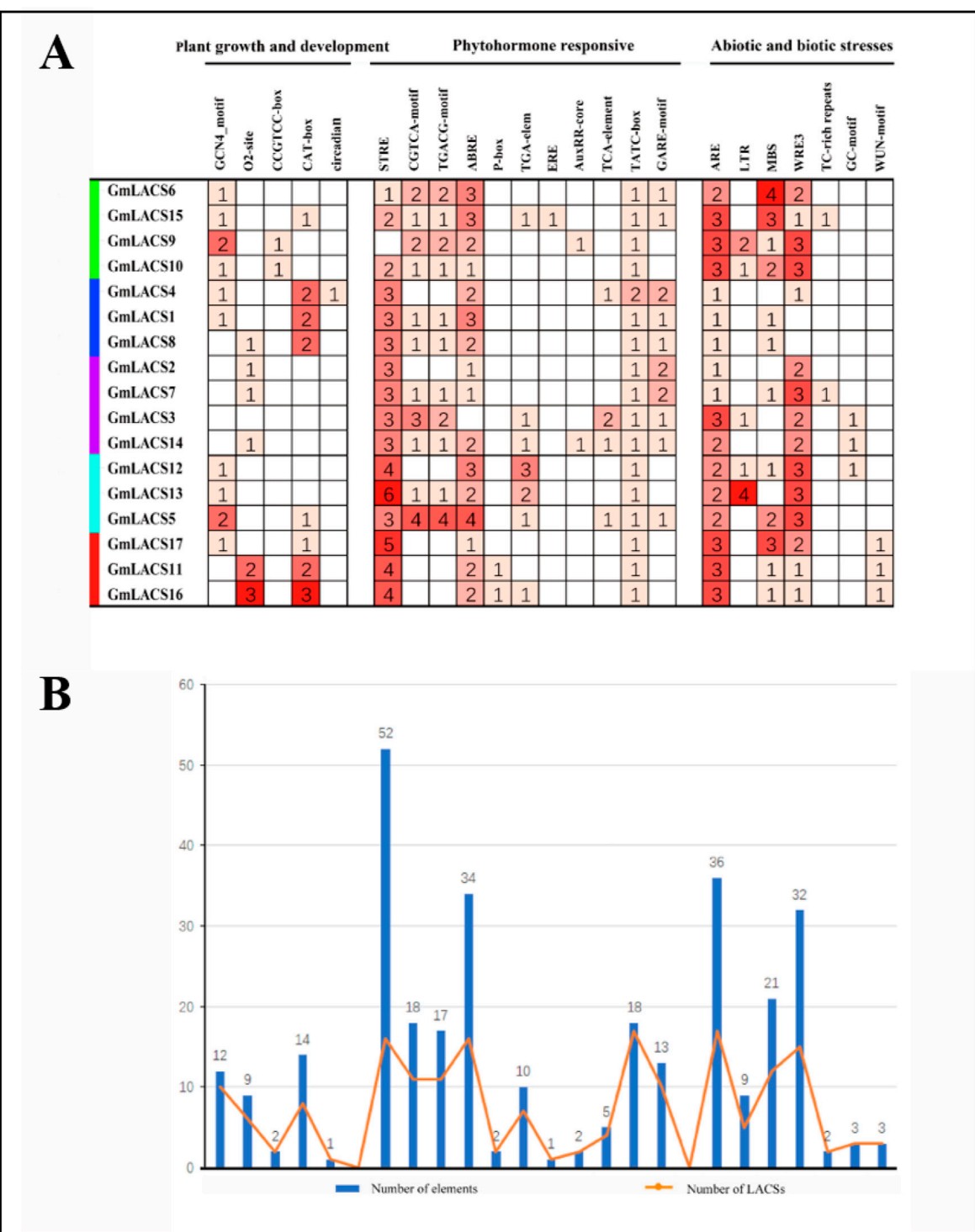

**Figure 4.** *Cis*-element analysis of 2kb promoter region upstream of *GmLACSs* start codon. (**A**) Number of each cis-acting element in the promoter region (2 kb upstream of the translation start site) of *GmLACS* genes. (**B**) Statistics for the total number of *GmLACS* genes, including the corresponding cis-acting element (orange) and the total number of cis-acting elements in the *GmLACS* gene family (blue box). On the basis of the functional annotation, the cis-acting elements were classified into three major classes: plant growth and development, phytohormone response, and abiotic and biotic stress response.

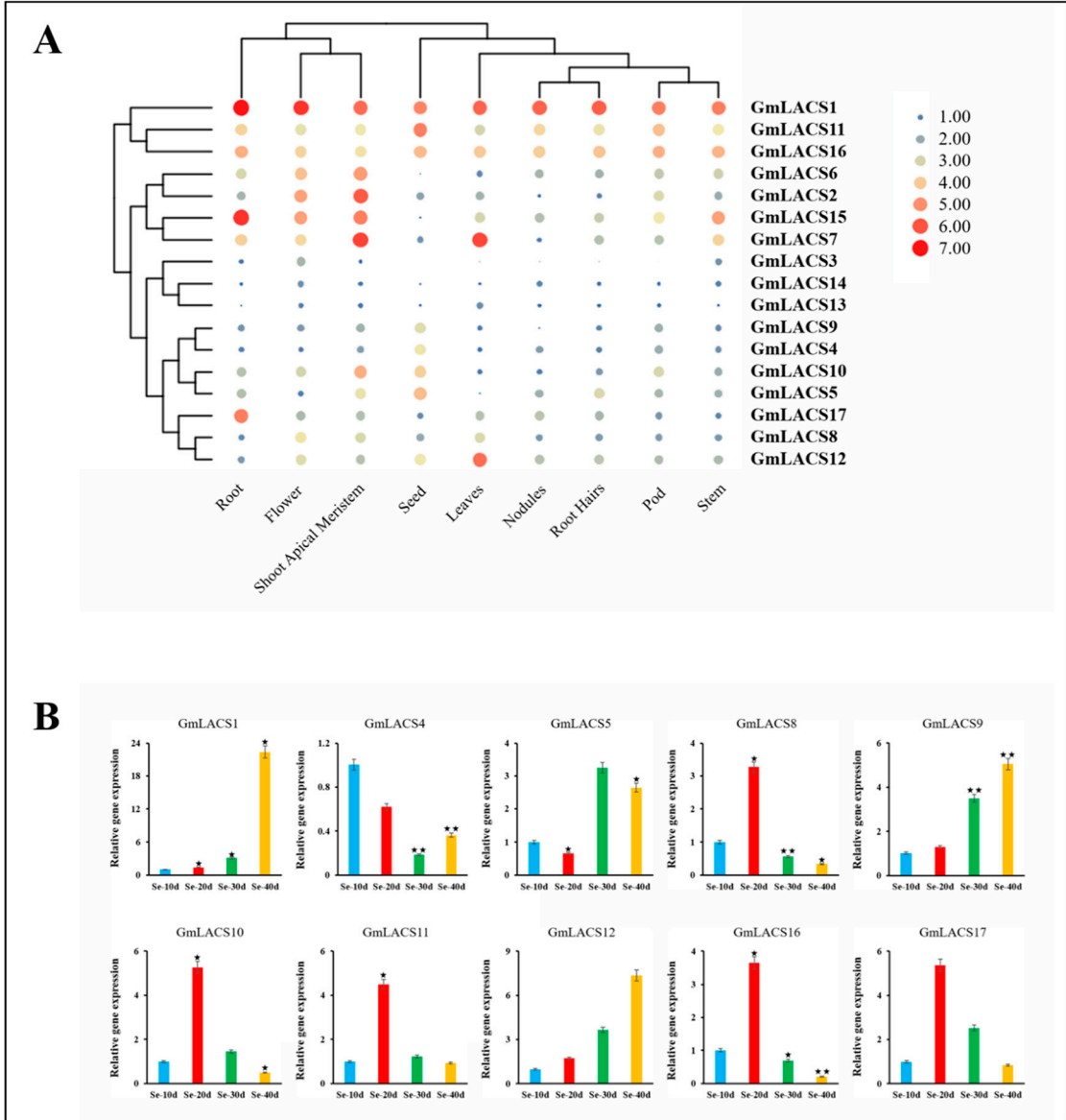

**Figure 5.** Expression profiles of *GmLACSs* in multiple tissues and developmental stages. (**A**) Expression patterns cluster analysis of *GmLACSs* involved in tissue development. The transcripts of *GmLACSs* genes in various tissues are investigated from Phytozome database. The results are shown as heat maps. The color scale represents log2 expression values, with red denoting high-level transcription and blue denoting low-level transcription. The size of the circle also indicates the transcription level, with a larger circle indicating high-abundance transcripts. (**B**) Transcriptome analysis was performed at 10, 20, 30 and 40 d (DAF) of soybean seed germination. The transcripts of *GmLACSs* at four DAFs were studied. Three biological replicates for each tissue were collected. Asterisks above bars denote a statistically significant difference by Student's *t*-test (* $p < 0.05$, ** $p < 0.01$).

### 3.7. Transcript Level of GmLACSs under Abiotic Stress

The *LACS* gene was known to be important for stress adaptation in several model plants. An analysis of the *GmLACSs* promoter revealed that many potential homeopathic elements and transcription-binding motifs (ABRE, TCA-element, ARE, MBS) were involved in response to abiotic stress, such as, northeast China often experiences low temperature and drought stress. Therefore, in order to further understand the response of *GmLACSs* to abiotic stress, we analyzed the transcription profile of *GmLACSs* under low temperature (4 °C), alkali (100 mM NaHCO₃), osmotic (20% PEG or 200 mM mannitol) and drought stress. As shown in the Figure 6, different GmLACS gene were significantly different under different

stress conditions. The expression levels of *GmLACS* genes (*GmLACS1/4/5/8/9/10/15/16/17*) were increased under all three stresses. The transcriptional expression of *GmLACS9* was over 75 times under alkali stress for 24 h, drought stress for 12 h, and over 50 times under low temperature stress for 12 h. The transcriptional expression of *GmLACS15* was more than 20 times under alkaline stress and drought stress for 6 h. The transcriptional expression of *GmLACS17* was more than 160 times under 12 h alkali stress. The expression levels of a few *GmLACS* genes (*GmLACS3/11/12/13/14*) were low under alkali stress and low temperature stress without significant change, and *GmLACS13* even showed negative regulation at all time periods. In addition, the expression levels of most *GmLACS* genes were significantly upregulated in drought stress, especially *GmLACS9* and *GmLACS17*, which were significantly increased at 12 h. However, the expression level of *GmLACSs* gene was significantly upregulated under 6h drought stress and was higher than that under alkali and low temperature stress under the same gene (Figure 6).

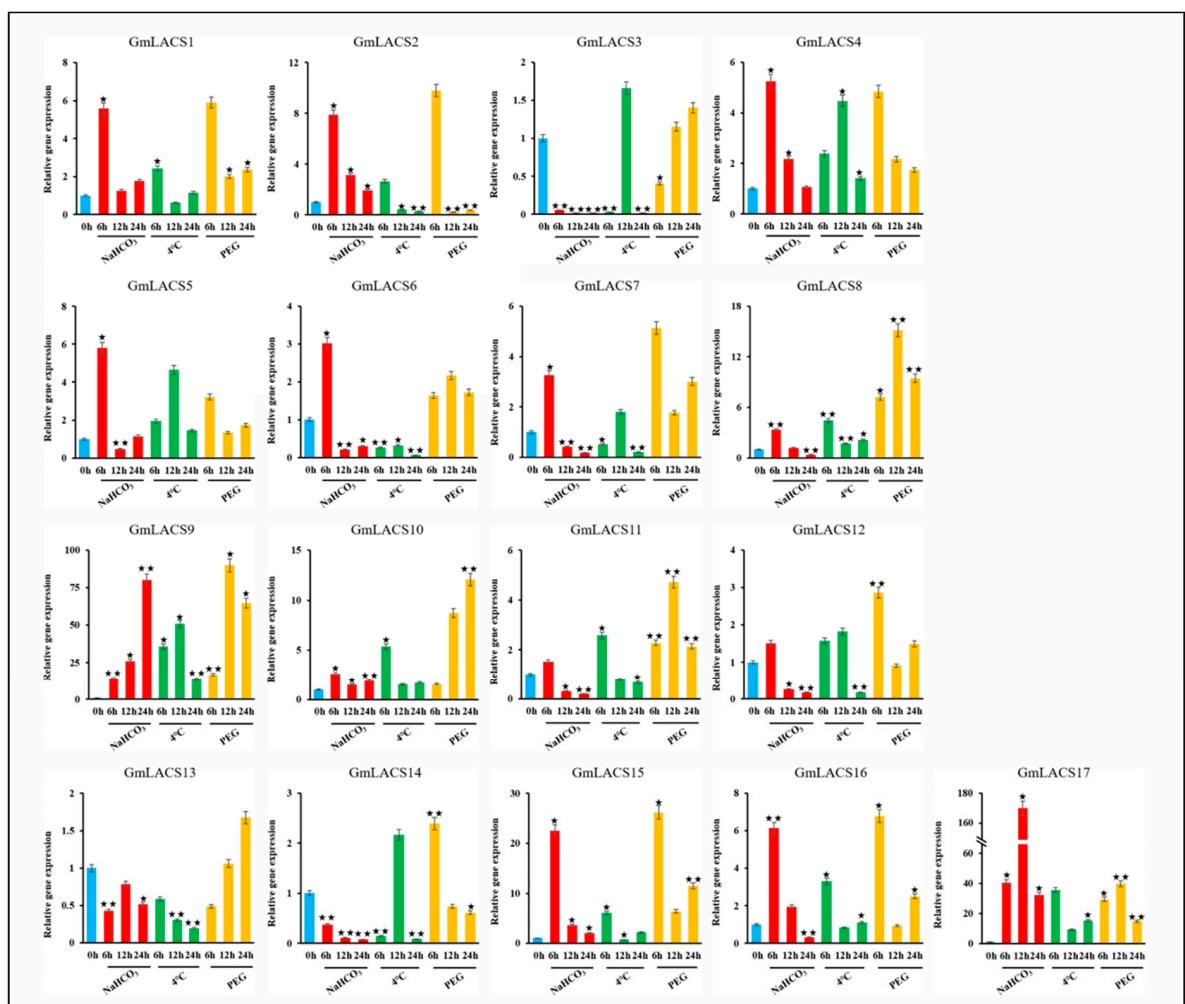

**Figure 6.** Expression profiles of *GmLACSs* at 4 °C, 100 mM NaHCO$_3$, 20% PEG and water (control) for 0, 6, 12 and 24h. The expression of *GmLACSs* in non-stress conditions was used as a calibrator. Three separate biological replicates, qRT-PCR results were analyzed by $2^{-\Delta\Delta ct}$ method. Asterisks on the bar chart indicate statistically significant differences in Student's *t*-test (* $p < 0.05$, ** $p < 0.01$).

### 3.8. Co-Expression Analysis of Transcription Factors and GmLACSs in Soybean

The significant responses of some soybean *GmLACS* genes aroused our interest and necessity to explore their regulation at the transcriptional level. Co-expression analysis of transcription factors (TFs) and highly responsive *LACSs* was performed using Cytoscape V 3.6.0. We used plants PlanTFDB TF database (http://planttfdb.cbi.pku.edu.cn/) (accessed

on 26 February 2022) from the drought transcriptome data screening of transcription factors. There are about 2700 transcription factors in the drought transcriptome of soybean, among which the most abundant TF families are bHLH, MYB, ERF, C2H2 and WRKY. We named the genes *GmLACS2/3/8/9/11/17* that strongly respond to drought as "guide" and described the co-expression relationship with corresponding transcription factors. As shown in Figure 7A, a total of 22 transcription factors were involved in the regulation of 6 guide genes, including bHLH (3), MYB (7), ERF (1), C2H2 (1) and WRKY (2), and a co-expression network was established between guide gene and corresponding transcription factors.

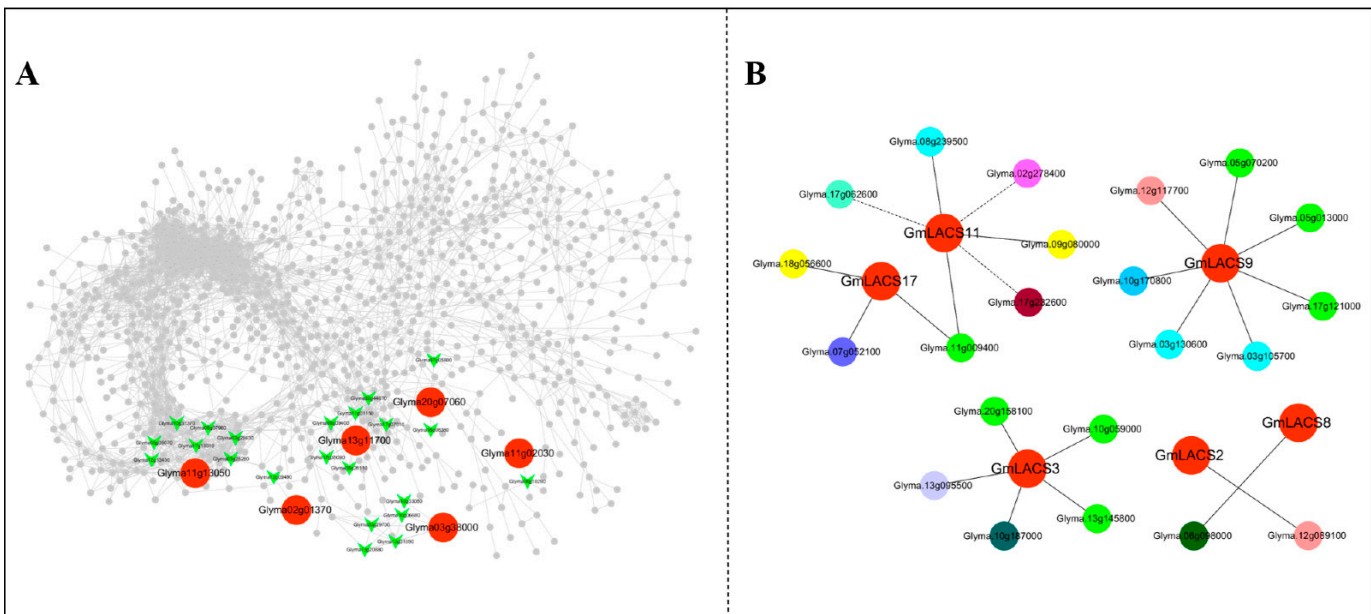

**Figure 7.** Co-expression network of transcription factors and *LACSs* in soybean. (**A**) Under drought stress; (**B**) Co-expression network of individual guide genes. When Pearson correlation is greater than 0.900, co-expression relationship is established, and a Perl script with the default value of 0.6 is used to build co-expression network. Different colored circles represent different transcription factors.

Meanwhile, we also constructed the co-expression network of individual guide genes (Figure 7B). One transcription factor was found to be significantly associated with the expression of *GmLACS2*, namely g2-like (Glyma.12G089100, r = 0.903). A total of 5 transcription factors were found to be significantly correlated with GmLACS3 expression, including *ERF* (*Glyma.10G187000*, r = 0.964), *MYB* (*Glyma.10G059000*, r = 0.905; *Glyma.13G145800*, r = 0.905; *Glyma.20G158100*, r = 0.944), *C2H2* (*Glyma.13G095500*, r = 0.905). Moreover, 1 transcription factor was found to be significantly associated with *GmLACS8* expression, namely *gATA* (*Glyma.06G098000*, r = 0.920). A total of 7 transcription factors were found to be significantly associated with *GmLACS9* expression, including *G2-like* (*Glyma.12G117700*, r = 0.935), *bHLH* (*Glyma.03G130600*, r = 0.923; *Glyma.03G105700*, r = 0.911), *MYB* (*Glyma.05G013000*, r = 0.910; *Glyma.05G070200*, r = 0.910; *Glyma.17G121000*, r = 0.914), *C3H* (*Glyma.10G170800*, r = 0.909). Furthermore, 6 transcription factors were found to be significantly associated with *GmLACS11* expression, including *MYB* (*Glyma.11G009400*, r = 0.947), AP2 (*Glyma.17G062600*, r = −0.933), *bHLH* (*Glyma.08G239500*, r = 0.930), *GRF* (*Glyma.17G232600*, r = −0.907), *WRKY* (*Glyma.09G080000*, r = 0.903), *HSF* (*Glyma.02G278400*, r = −0.900). Finally, 3 transcription factors were found to be significantly associated with *GmLACS17* expression, including *HD-ZIP* (*Glyma.07G052100*, r = 0.942), *WRKY* (*Glyma.18G056600*, r = 0.911), *MYB* (*Glyma.11G009400*, r = 0.947). Therefore, these transcription factors may regulate the response of six strongly responsive genes to drought stress.

## 4. Discussion

Soybean is a major source of plant proteins and oils, providing more than a quarter of the world's protein for food and animal feed [52–54]. *LACS* genes recently have been obtained from Arabidopsis, apricot, upland cotton, and brassica napus, and their structure and functional roles have been preliminarily studied [20,31]. Although *LACS* genes played an important role in oil synthesis and stress resistance of plants. So far, no detailed identification and functional studies of this family of genes have been reported in soybean. In this study, the LACS family genes of soybean were studied in detail. The expression characteristics of LACS family genes in different tissues of soybean and in response to abiotic stress were quantitatively verified. The drought-responsive gene GmLACS2/3/8/911/17 of this gene family was co-expressed. This study provides a new direction and idea for understanding the role of soybean LACS in plant growth and development.

In this study, the *GmLACS* gene family was identified and analyzed based on the annotated information of the genome and the comparison results of *LACS* homologous genes in *Arabidopsis thaliana*. A total of 17 members of soybean *LACS* gene family were screened from the online database and named *GmLACS1-17*, respectively (Table 1). This result is higher than of *Arabidopsis* (9), rape (12) and cabbage (16), but less than that of Brassica napus (34) [20,38]. These differences indicate that the number of genes in plants is related to genome size and gene replication events [3]. Similarly to the LACS results of the studied plants including *Arabidopsis thaliana*, all of the GmLACS proteins contain AMP-binding domain (PF00501) (Figure 1B). The predicted number and sequence of conservative motifs of all GmLACS proteins are basically the same (Figure 1C), so GmLACSs are highly conserved amino acid sequences [55]. In order to further clarify the relationship between members of soybean LACS family and LACS family coding genes from other plants, the screened LACS proteins of soybean, *Arabidopsis*, maize and upland cotton were analyzed. The results show (Figure 1A) that all of the LACS involved in the comparison originated from a common ancestor. In combination with the classification of the existing nine *Arabidopsis* LACS [21,24], all of the LACS involved in the comparison are also divided into subfamilies, and the GmLACS gene structure of each subfamily is the same (Figure 2).

Compared with Arabidopsis thaliana, the homologous genes of all species have been widely extended. Among them, GmLACS5/12/13 and AtLACS9 are the same subfamily (Figure 1A). AtLACS9 is a widely studied member of the AtLACS family. Since it is located in the de novo synthesis and plastid of plant fatty acids, it is considered to be the main LACS subtype formed by acyl-CoA and participates in tag biosynthesis [21,24]. Therefore, GmLACS5/12/13, which is homologous to AtLACS9, may be the main gene of soybean LACS gene family that produces tag in the long-chain fatty acid synthesis pathway. Similarly, GmLACS 6/9/10/15 and AtLACS2 are the same subfamily (Figure 1A). Ayaz et al. [44] found that *AtLACS2* has a high similarity with the four *GmLACS* (the same as the genes in this study). Arabidopsis homologous gene AtLACS2 plays a key role in cutin and wax synthesis, and its transcripts are highly accumulated in elongated tissues. The rape homologous gene BnLACS2 participates in seed growth and accumulates during seed development, during which tag will be actively produced [35]. The apple homologous gene MdLACS2 has been proved to be involved in the synthesis of epidermal wax and is highly expressed in the peel [56]. Therefore, GmLACS6/9/10/15 is also considered to be involved in the synthesis of soybean cuticle. GmLACS1/4/8 and AtLACS3/4/5 are the same subfamily (Figure 1A). In Arabidopsis, AtLACS4 and AtLACS1 are partially redundant in providing substrates for stem and leaf cuticle wax biosynthesis and pollen coat lipid formation [20]. AtLACS3 may be strongly expressed in the stem epidermis [22]. AtLACS5 is expressed in anthers. MdLACS4, an apple homologous gene, induced early flowering of Arabidopsis thaliana and enhanced its ability to resist abiotic stresses [56]. Therefore, GmLACS1/4/8 may be related to soybean flowering. Combined with other related studies, it is speculated that other soybean LACS also have similar physiological functions with other homologous LACS genes. These results provide a direction for the detailed study of



the mechanism of soybean LACS gene participating in fatty acid metabolism in different organelles and provide a basis for the functional overlap among the members of the family.

In addition, Xiao et al. [31] found that the length of LACS protein in brassica napus was 129–960 aa, and the theoretical isoelectric point was 5.11–9.15. Zhang et al. [30] found that the length of LACS protein in apple was 596–730 aa, the molecular weight was 64.96–79.44 KDa, and the theoretical isoelectric point was 5.56–8.12. In this study, the length of GmLACS protein was 648–726 aa, the theoretical isoelectric point was 5.58–8.24, and the molecular weight was 70,974.48–79,550.53 Da (Table 1). Compared with apple and brassica napus, the protein length and molecular weight of GmLACS showed no significant difference and were neutral. It was confirmed that LACS proteins from different plants had similar physical and chemical properties, and also verified the hypothesis that LACS proteins had similar functions in this respect.

The expression patterns of *LACS* gene in Arabidopsis thaliana and soybean showed different expression patterns. In Arabidopsis thaliana, the expression levels of most *AtLACS* greatly genes vary during flower development, and the expression levels of *AtLACS1/2/4/6/8/9* were higher than those of *AtLACS3/5/7* [20]. In this study, *GmLACSs* also showed different expression patterns in different tissues. It was noteworthy that the transcription level of *GmLACS1* in all tissues was relatively high (Figure 5), which was similar to the corresponding gene *AtLACS4* in Arabidopsis and the apple homologous gene *MdLACS4* [20]. The expression level of *GmLACS7* was relatively high in apex meristem and leaf, so the expression level of their homologous gene *AtLACS1* was also relatively high in apex meristem and leaf [20]. These results suggested that *LACS* gene functions conserved in organs of different species. Therefore, we speculated that *GmLACS5/12/13* might play an important role in lipid metabolism, similarly to *AtLACS9*, a homolog of *Arabidopsis* thaliana [24,57].

In addition to different differential expressions affecting plant growth, the important role of *LACS* gene in the stress response mechanism had also been widely confirmed [58–60]. We identified 23 cis acting elements related to plant growth and development, plant hormones, abiotic and biological stresses. As shown in Figure 4A, seven of them are related to abiotic and biological stresses. These elements may be involved in the response of plants to drought, osmotic, alkaline, low temperature and other stresses (Figure 4). In order to further determine the significant difference of *GmLACS* under low temperature, drought and alkali stress, we conducted a transcription analysis (Figure 6). The results showed that the expression of *GmLACS9/15/17* was significantly increased under drought stress, which was consistent with the reported hypersensitivity to drought of several Arabidopsis homologous single mutant atlacs2 and double mutant atlacs1 atlacs2. In addition, due to functional redundancy, higher-order Arabidopsis atlacs mutants often show higher sensitivity to drought. For example, the three mutants atlacs1 atlacs2 atlacs4 are a good example. In apples, the expression of MdLACS2 and MdLACS4 can reduce epidermal permeability, reduce water loss, and enhance the resistance to drought and salt stress [25]. Finally, in order to understand the role of transcription factors in the regulation of gene expression under drought stress, we analyzed the co expression of transcription factors of the most sensitive lacs gene. Under drought stress, the expression of *bHLH*, *MYB*, *ERF*, *C2H2* and *WRKY* were significantly correlated with the expressions of *GmLACS2/3/8/9/11/17* under drought stress (Figure 7), revealing the regulatory role of soybean *LACS* gene at the transcription level. The co-expression analysis of transcription factors and soybean *LACS* genes provided preliminary information for understanding the regulatory mechanism of soybean *LACS* under drought stress, but the transcriptional regulatory network needs to be further studied. These results show that LACSs are indispensable in the process of plant drought resistance.

## 5. Conclusions

In summary, a total of 17 soybean *LACS* genes were categorized into 6 distinct clusters. According to the correlation between these genes and the corresponding transcription

factors, six of the genes of the drought reflected strong results of the drought, and the network was established. However, a clearer picture of the molecular function requires further research.

**Supplementary Materials:** The following supporting information can be downloaded at: https://www.mdpi.com/article/10.3390/agronomy12071496/s1. Figure S1: Chromosome distribution of LACS genes in soybean. Table S1: All primers used for qRT-PCR analysis. Table S2: The gene ID of LACS genes used in this study. Table S3: Protein physicochemical properties and supplementary information of GmLACSs. Table S4: Intraspecific homologous pairs of soybean.

**Author Contributions:** Conceptualization and methodology, J.W., X.L. and X.Z. (Xunchao Zhao); investigation, C.N., H.L. and H.M.; data curation, J.Z., J.X. and X.Z. (Xunchao Zhao); writing—original draft preparation, X.Z. (Xue Zhao) and Y.H.; writing—review and editing, J.W., X.Z. (Xue Zhao) and Y.H.; writing; discussing and revising, J.W., X.Z. (Xue Zhao) and Y.H. All authors have read and agreed to the published version of the manuscript.

**Funding:** This study was financially supported by the Chinese National Natural Science Foundation (31871650, 31971967), National Key R & D Project (2021YFD1201604, 2021YFF1001204), the National Project (2014BAD22B01, 2016ZX08004001-007), the Youth Leading Talent Project of the Ministry of Science and Technology in China (2015RA228), The National Ten-thousand Talents Program, The national project (CARS-04-PS04).

**Institutional Review Board Statement:** Not applicable.

**Informed Consent Statement:** Not applicable.

**Data Availability Statement:** The data that support this study are available in the article and accompanying online supplementary material.

**Conflicts of Interest:** The authors declare no conflict of interest.

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
