# Peer review of "Genome-Wide Identification and Characterization of the Abiotic-Stress-Responsive LACS Gene Family in Soybean (Glycine max)"

_agronomy, doi:10.3390/agronomy12071496_

Round 1
Reviewer 1 Report
Overall comment(s)
This work carries significant scientific merit and could potentially be publishable upon some major revisions suggested. The authors do an impressive job of characterizing an important gene family controlling stress response in higher plants but fall short of drawing showing the relevance of such work to the scientific community.
Major issues
L 88-91: The authors set a fairly laxed BLASTP threshold of E-value (<10−10) and a sequence identity threshold > 90%. Why not a more stringent threshold? Could you provide a justification for this choice?
L 92-94: Please provide a description for candidate validation
Methods: Critical detail of data processing, qc, analysis are missing throughout the manuscript.
Minor issues
L12, 13, 33, etc., These are established facts in science. Please change to present tense i.e., LACSs “are” not “were” …, FAs “are” not “were”., Please correct throughout the manuscript.
Author Response
Dear Editor and Reviewer,
On behalf of all the authors, we thank you very much for the help in reviewing our manuscript entitled “Genome-Wide Identification and Characterization of the Abiotic-Stress-Responsive LACS Gene Family in Soybean (Glycine max)”. We are very grateful to the constructive comments from all reviewers and efforts made by the editor. We have revised all parts in the main text required by the reviewers. The point by point for revision was listed below. We hope these revisions would meet your requirements.
Many thanks and best regards,
Yingpeng Han
Soybean Research Institute
Northeast Agricultural University
Harbin, China 150030
We respond to the Reviewers’ Comments in order below:
Reviewer #1:
Overall comment(s):
Q: This work carries significant scientific merit and could potentially be publishable upon some major revisions suggested. The authors do an impressive job of characterizing an important gene family controlling stress response in higher plants but fall short of drawing showing the relevance of such work to the scientific community.
Response: Thank you very much for your time on reviewing our manuscript conscientiously. We have described this problem in the new discussion. (Page17, line 561-594)
Major issues:
Q: L88-91: The authors set a fairly laxed BLASTP threshold of E-value (<10−10) and a sequence identity threshold >90%. Why not a more stringent threshold? Could you provide a justification for this choice?
Response: Thank you for pointing out this problem. We chose BLASTP threshold of E-value (<10−10) and a sequence identity threshold >90% based on the decision made by querying relevant literature.
To obtain all G6PDHs from the soybean genome, a systematic BLASTP search was carried out against the soybean genetics and genomics database (SoyBase) using the published A. thaliana G6PDHs as queries. The protein sequences of putative soybean G6PDH family members with an E-value of <10−10 and a sequence identity threshold >90% were downloaded. (Zhao, Y.; Cui, Y.; Huang, S.; Yu, J.; Wang, X.; Xin, D.; Li, X.; Liu, Y.; Dai, Y.; Qi, Z.; Chen, Q. Genome-Wide Analysis of the Glucose-6-Phosphate Dehydrogenase Family in Soybean and Functional Identification of GmG6PDH2 Involvement in Salt Stress. Front Plant Sci 2020, 11, 214.)
The predicted peptide sequences of AQP members were acquired from the genome database (unpublished) of kernel-using apricot (P. armeniaca L.), also known as Longwangmao (Chinese Pinyin name), to construct a local protein database. BLASTP searches were performed using AQP protein sequences of A. thaliana and poplar as queries in The Arabidopsis Information Resource (TAIR) 1 and the Phytozome database (release version 12.02) as previously described, with an E-value of 1e−10 and a minimum amino acid identity of 50%. (Li, S.; Wang, L.; Zhang, Y.; Zhu, G.; Zhu, X.; Xia, Y.; Li, J.; Gao, X.; Wang, S.; Zhang, J.; Mo, W. Genome-Wide Identification and Function of Aquaporin Genes During Dormancy and Sprouting Periods of Kernel-Using Apricot (Prunus armeniaca L.). Front Plant Sci 2021, 12, 690040.)
Q: L92-94: Please provide a description for candidate validation.
Response: Thank you for pointing out this problem. According to the protein sequence of the screened gene, query in the genomic mode of the domain database (SMART) to show whether its domain is AMP-binding domain (PF00501) to determine whether it is a candidate gene. EMBL-PFAM is a database of protein families, including annotations and multiple sequence alignments generated using hidden Markov models. Enter PF00501 into PFAM-A sub library, and the query results show that its domain belongs to LACS family. It is further judged that the candidate genes belong to LACS family genes. (Page3, line 137-144)
Q: Methods: Critical detail of data processing, qc, analysis are missing throughout the manuscript.
Response: Dear reviewer, thank you for the nice suggestion. We have revised the full text, including abstract, preface, discussion, some results and some methods.
Minor issues:
Q: L12, 13, 33, etc., These are established facts in science. Please change to present tense i.e., LACSs “are” not “were” …, FAs “are” not “were”., Please correct throughout the manuscript.
Response: Dear reviewer, thank you for the nice suggestion. We have revised the whole manuscript, hoping to solve this problem. For example, line 12, LACSs "are"; Line 41, FAs "are"; Line 48, acids "are"; Line 53, which "are".

Reviewer 2 Report
This is a catalog of LACS genes in soybean. While I understand the need to do this work, I think this description has not gone very far to draw out the parallels with the Arabidopsis family. For this reason, I wonder if this paper merits publication. I know this work is needed, but very few conclusions about what these genes do in soybeans have been made. I think more inferences can be made about the correlations between Arabidopsis and soybean orthologs.
Why are all of the soybean LACS expected to localize to the peroxisome? Is this true of Arabidopsis? This observation doesn't seem consistent with the description in the introduction. I think a longer discussion of this is merited.
Specific issues the need to be addressed
- The sentence from line 43-45 doesn't make sense and I can't figure out what the authors are trying to say.
-
Line 190: Discussing MW of DNA, I think you should be referring to protein MW.
- Line 173: BlasP was used. I think the authors mean BlastP.
- I don’t understand what Figure 4B is showing
Author Response
Dear Editor and Reviewer,
On behalf of all the authors, we thank you very much for the help in reviewing our manuscript entitled “Genome-Wide Identification and Characterization of the Abiotic-Stress-Responsive LACS Gene Family in Soybean (Glycine max)”. We are very grateful to the constructive comments from all reviewers and efforts made by the editor. We have revised all parts in the main text required by the reviewers. The point by point for revision was listed below. We hope these revisions would meet your requirements.
Many thanks and best regards,
Yingpeng Han
Soybean Research Institute
Northeast Agricultural University
Harbin, China 150030
We respond to the Reviewers’ Comments in order below:
Reviewer #2:
Q: This is a catalog of LACS genes in soybean. While I understand the need to do this work, I think this description has not gone very far to draw out the parallels with the Arabidopsis family. For this reason, I wonder if this paper merits publication. I know this work is needed, but very few conclusions about what these genes do in soybeans have been made. I think more inferences can be made about the correlations between Arabidopsis and soybean orthologs.
Response: Dear reviewer, thank you for the nice suggestion. We have rewritten the preface and discussion to make this study have more connections and analogies with Arabidopsis and draw more conclusions. (Page1, line 40; Page15, line 466)
Q: Why are all of the soybean LACS expected to localize to the peroxisome? Is this true of Arabidopsis? This observation doesn't seem consistent with the description in the introduction. I think a longer discussion of this is merited.
Response: Thank you for pointing out this. We use cell ploc 2.0 online website(http://www.csbio.sjtu.edu.cn/bioinf/Cell-PLoc-2/)Subcellular localization of all GmLACS protein sequences was predicted, but the results did not seem very satisfactory. All the predicted results showed that they were localized in peroxisome. This is obviously inconsistent with the subcellular localization of Arabidopsis homologous genes. Therefore, we deleted the relevant contents in this study. For the subcellular localization of GmLACS, we also think this experiment is necessary. We want to try other methods to get results, which will continue to be completed in our follow-up work. We hope to have the opportunity to cooperate with you on this subject.
Specific issues the need to be addressed:
Q: The sentence from line 43-45 doesn't make sense and I can't figure out what the authors are trying to say.
Response: Dear reviewer, thank you for the nice suggestion. This sentence has been deleted, and the rest of this paragraph has been rewritten.
Q: Line 190: Discussing MW of DNA, I think you should be referring to protein MW.
Response: Dear reviewer, thank you for the nice suggestion. Your suggestion has been accepted. Double stranded DNA has been changed to protein at line 242.
Q: Line 173: BlasP was used. I think the authors mean BlastP.
Response: Thank you for pointing out this. In line 224, “BlasP” has been replaced by “BlastP”.
Q: I don’t understand what Figure 4B is showing.
Response: Thank you for pointing out this problem. In order to get this conclusion, biotic and abiotic stresses play an important role in the functional regulation of plant growth and development. We analyzed the promoter cis acting elements of all GmLACS gene sequences, obtained the sum of the number of each cis acting element in the promoter region of GmLACS gene (2kb upstream of the translation start site), compared the number of elements related to plant growth and development, plant hormone response and biological and abiotic stress, and found that there were more elements related to biological and abiotic stress. Therefore, Figure 4B is intended to illustrate this phenomenon.

Reviewer 3 Report
Agronomy Review Report - -
General comments:
In general, the manuscript titled ) has a valuable topic. the work a significant contribution to the field. The manuscript is well written. the work scientifically sound and not misleading The English language and style are fine and readable except for some English language check required.
Are there appropriate and adequate references to related and previous work
There are some minor comments.
Detailed comments:
Title:
Keywords:
Please add Resistance to the keywords list.
Abstract:
The aim of the study and the main objectives were not clearly stated.
Please state the aim of this study clearly in this section.
The work well organized and comprehensively described
Introduction:
This section didn’t provide enough background about the topic. The introduction needs to be elongated and enriched.
Materials and Methods:
The experimental design is adequate and suitable to the current study.
Results:
The results were well presented BUT with a poor discussion.
Discussion:
This section is poorly written
As mentioned before, this section is poorly written. I had a hard time to relate the discussion section with the corresponding data in the tables and the figures. There are 7 figures and one table that hardly mentioned and poorly discussed.
*Please rewrite this section and provide the appropriate citations in argument, and valuable discussion to the current results.
*For best discussion to the provided data the author is strongly advised to combine the results section and the discussion section.
Conclusion:
This section is ok. This section provides a good conclusion for the study and includes the significant findings with some recommendations for further study about this point.
References:
The authors provided enough citations, and it was UpToDate.
Author Response
Dear Editor and Reviewers,
On behalf of all the authors, we thank you very much for the help in reviewing our manuscript entitled “Genome-Wide Identification and Characterization of the Abiotic-Stress-Responsive LACS Gene Family in Soybean (Glycine max)”. We are very grateful to the constructive comments from all reviewers and efforts made by the editor. We have revised all parts in the main text required by the reviewers. The point by point for revision was listed below. We hope these revisions would meet your requirements.
Many thanks and best regards,
Yingpeng Han
Soybean Research Institute
Northeast Agricultural University
Harbin, China 150030
We respond to the Reviewers’ Comments in order below:
Reviewer #3:
General comments:
Q: In general, the manuscript titled () has a valuable topic. the work a significant contribution to the field. The manuscript is well written. the work scientifically sound and not misleading The English language and style are fine and readable except for some English language check required. Are there appropriate and adequate references to related and previous work. There are some minor comments.
Response: Thanks for your nice comment.
Detailed comments:
Keywords:
Q: Please add Resistance to the keywords list.
Response: Dear reviewer, thank you for the nice suggestion. Resistance has been added as a keyword. (Page1, line 38)
Abstract:
Q: The aim of the study and the main objectives were not clearly stated. Please state the aim of this study clearly in this section.
Response: Dear reviewer, thank you for pointing out this. The purpose and main objectives of the research have been clarified and have been added to the beginning and the end of Abstract of the revised manuscript, respectively: “Long chain acyl CoA synthetases (LACSs) are a key factor for the formation of acyl-CoA after fatty acid hydrolysis, and play an important role in plant stress resistance. This gene family has not been research in soybeans. In this study, soybean (Glycine max (L.) merr.) whole genome was identified, the LACS family genes of soybean were screened, and the bioinformatics, tissue expression, abiotic stress, drought stress and co-expression of transcription factors of the gene family were analyzed to preliminarily clarify the function of the LACS family genes of soybean.” (Page1, line 12-19)
“These results provide a theoretical and empirical basis for clarifying the function of LACS family genes and abiotic stress response mechanism of soybean.” (Page1, line 34-36)
Q: The work well organized and comprehensively described.
Response: Thanks for your nice comment.
Introduction:
Q: This section didn’t provide enough background about the topic. The introduction needs to be elongated and enriched.
Response: Thank you very much for your suggestion. We have added new background content to the new preface, which fully enriches the introduction.
“FAs can be divided into three categories according to the length of their carbon chains, namely long-chain fatty acids (LCFA), medium chain fatty acids (MCFA) and short chain fatty acids (SCFA). The length of fatty acids in higher plants is usually between 14-20 carbon long-chain fatty acids. Acyl CoA synthase (ACS) is also known as fatty acid CoA ligase. According to the difference of carbon chain length of specific fatty acid substrates, ACS can be divided into the following four categories: super long chain (>C20), long chain (C14-C18), medium chain (C10-C12) and short chain (C6-C8) acyl CoA synthetases.” (Page2, line 56-62)
“The spatial distribution of LACS enzymes in cells is a factor that leads fatty acids to a specific metabolic fate. Consistent with this, in most eukaryotes LACS is encoded by different gene subfamilies in specific pathways, such as tissue-specific expression and subcellular location. However, LACS activity often shows significant overlap in substrate specificity, such as human fatty acid translocation, and this is also the case in Arabidopsis thaliana.” (Page2, line 70-75)
“In Arabidopsis thaliana, nine LACS isoforms were identified, which had different expression patterns and functions. In vitro enzyme activity analysis showed that all LACSs can effectively activate a variety of substrates. Meanwhile, most of the nine LACS genes in Arabidopsis have been isolated and mutant. The identification of these mutants and the analysis of subcellular localization of expression patterns revealed a complex LACS activity network involving different aspects of lipid metabolism.” (Page2, line 79-84)
“Among them, several LACS subtypes located in the endoplasmic reticulum can activate fatty acids to produce surface lipids. Long chain specificity analysis and the phenotype of lacs1 mutant showed that LACS1 played a major role in the production of long chain acyl-CoA, and LACS1 was the precursor of cuticle wax. Together with LACS1, LACS2 activates VLCFAs to produce wax components and to bind to keratin in C16 and C18 acyl groups. LACS4 and LACS1 are partially redundant in providing substrates for wax biosynthesis in stem and leaf cuticle and lipid formation in pollen coat. LACS3 may be strongly expressed in stem epidermis, but it has not been studied so far. LACS5 is expressed in anthers.” (Page2, line 85-93)
“Similarly, the identification of lacs6 and lacs7 double mutants showed that, LACS6 and LACS7 were involved in the degradation of fatty acids in peroxidase. In addition, LACS9 was considered to be the main LACS subtype involved in the formation of acyl-CoA. Participate in TAG biosynthesis. LACS9 and LACS4 overlap with LACS8 in function. The destruction of LACS 8 under the background of lacs9 and lacs4 may lead to lethality.” (Page2, line 95-100)
Discussion:
Q: As mentioned before, this section is poorly written. I had a hard time to relate the discussion section with the corresponding data in the tables and the figures. There are 7 figures and one table that hardly mentioned and poorly discussed.
*Please rewrite this section and provide the appropriate citations in argument, and valuable discussion to the current results.
*For best discussion to the provided data the author is strongly advised to combine the results section and the discussion section.
Response: Dear reviewer, thank you for the nice suggestion. We have rewritten this part in the new manuscript, making the discussion more closely related to the results, and making some sufficient references. (Page15, line 466)
Conclusion:
Q: This section is ok. This section provides a good conclusion for the study and includes the significant findings with some recommendations for further study about this point.
Response: Thanks for your nice comment.
References:
Q: The authors provided enough citations, and it was UpToDate.
Response: Thanks for your nice comment.

Round 2
Reviewer 2 Report
The concerns and issues I raised previously have been addressed. I have no further issues with this submission.